# Diagnostic Reference Levels in Interventional Pediatric Cardiology: Two-Year Experience in a Tertiary Referral Hospital in Latin America

**DOI:** 10.3390/children10101588

**Published:** 2023-09-23

**Authors:** Ana M. Aristizabal, Jaiber A. Gutiérrez, Emmanuel Ramirez, Valentina Mejía-Quiñones, Carlos Ubeda, Walter Fernando Mosquera, Walter Mosquera

**Affiliations:** 1Servicio de Cardiología Pediátrica, Departamento Materno Infantil, Fundación Valle del Lili, Cra. 98 No. 18–49, Cali 760032, Colombia; jaiber.gutierrez@fvl.org.co (J.A.G.); walter.mosquera@fvl.org.co (W.M.); 2Facultad de Ciencias de la Salud, Universidad Icesi, Cali 760031, Colombia; valentina.mejia@fvl.org.co; 3Oficina de Física Médica & Protección Radiológica, Fundación Valle del Lili, Cra. 98 No. 18–49, Cali 760032, Colombia; emmanuel.ramirez@fvl.org.co; 4Centro de Investigaciones Clínicas, Fundación Valle del Lili, Cra. 98 No. 18–49, Cali 760032, Colombia; 5Diagnóstico por Imagen, Laboratorio de Dosimetría Personal (LABODOP), Departamento de Tecnología Médica, Facultad de Ciencias de la Salud, Universidad de Tarapacá, Arica 1020000, Chile; cubeda@academicos.uta.cl; 6Servicio de Hospitalización, Fundación Valle del Lili, Cra. 98 No. 18–49, Cali 760032, Colombia; walmos94@gmail.com

**Keywords:** pediatric cardiac catheterization, radiation dose, air kerma-area product, body weight

## Abstract

The goal of the present study was to propose the first local diagnostic reference levels (DRLs) for interventional pediatric cardiology procedures in a large hospital in Colombia. The data collection period was from April 2020 to July 2022. The local DRLs were calculated as the 3rd quartile of patient-dose distributions for the kerma-area product (*P*_ka_) values. The sample of collected clinical procedures (255) was divided into diagnostic and therapeutic procedures and grouped into five weight and five age bands. The *P*_ka_ differences found between diagnostic and therapeutic procedures were statistically significant in all weight and age bands, except for the 1–5-year age group. The local DRLs for weight bands were 3.82 Gy·cm^2^ (<5 kg), 7.39 Gy·cm^2^ (5–<15 kg), 19.72 Gy·cm^2^ (15–<30 kg), 28.99 Gy·cm^2^ (30–<50 kg), and 81.71 Gy·cm^2^ (50–<80 kg), respectively. For age bands, the DRLs were 3.97 Gy·cm^2^ (<1 y), 9.94 Gy·cm^2^ (1–<5 y), 20.82 Gy·cm^2^ (5–<10 y), 58.00 Gy·cm^2^ (10–<16 y), and 31.56 Gy·cm^2^ (<16 y), respectively. In conclusion, when comparing our results with other existing DRL values, we found that they are similar to other centers and thus there is scope to continue optimizing the radiation dose values. This will contribute to establishing national DRLs for Colombia in the near future.

## 1. Introduction

The development of cardiology has led to an increase in the number of interventional procedures. In recent decades, interventional pediatric cardiology (IPC) has transitioned from being primarily a diagnostic instrument to being a therapeutic approach that has markedly enhanced the prognosis of congenital heart defects [1,2].

However, it is known that these procedures may involve high doses of radiation, which increases the potential risk of stochastic effects, especially in pediatric patients, given the higher sensitivity of their tissues compared to adults [3,4]. 

Therefore, radiation protection strategies exist to avoid the deterministic effects and reduce the probability of stochastic effects as much as possible [3,5]. These strategies is embodied in three principles, that are part of the radiation protection system: (1) justification of the study by being clear that the exposure produces a net benefit versus the potential risks; (2) limitation of the dose to avoid exceeding the established values (only applicable to personnel and the general public, no dose limits are applied to patients); and (3) optimization to achieve doses as low as reasonably achievable, called the “ALARA” concept [5,6].

To assist in the optimization process related with patient radiation exposure, one of the main tools available is the use of diagnostic reference levels (DRLs) [7]. According to report 135 by the International Commission on Radiological Protection (ICRP), DRLs are a form of investigation level used for diagnostic and interventional procedures. They are used in medical imaging with ionizing radiation to indicate whether, under routine conditions, the amount of radiation used for a specified procedure is unusually high or low for that procedure. This last report introduced the terms “DRL quantity”, “DRL value”, “local DRL or typical value” and recommended using a facility’s median value (rather than mean value) for DRL quantity, given that this is recognized as more robust and representative for the patient population. Also, it is correct to use the third (3rd) quartile values as DRLs for local, national, or regional DRLs [8].

The ICRP has recommended using DRLs for fluoroscopy-guided procedures in all institutions, including IPC procedures [3,8]. Ideally, each institution should be able to establish its own values and each country should do the same; finally, these values should be grouped for each region of the world to compare them with each other. This will make it possible to identify whether the values obtained in a specific institution are higher or lower than the existing DRL and, based on these results, to assess the need to take corrective measures. 

Incipient efforts have been undertaken in our region to advance the establishment of DRLs [7,9,10]; however, according to our knowledge, in Colombia we only have the record by Mosquera et al., 2014 [11]. Therefore, this research sought to obtain the first local DRLs in our institution to perform a self-assessment and optimization of radiation protocols, in addition to contributing to the establishment of national DRLs for IPC. 

This study was carried out within the framework of the initiative “Optimization of Protection in Pediatric Interventional Radiology in Latin America and the Caribbean” (OPRIPALC), instituted in 2018. This program is a collaborative endeavor by the Pan-American Health Organization (PAHO) and the World Health Organization (WHO), in association with the International Atomic Energy Agency (IAEA), to assist their member countries in guaranteeing that the radiation exposure of pediatric patients is kept to a bare minimum during fluoroscopy-aided interventional procedures [12,13].

## 2. Materials and Methods

The observational study design was retrospective case series [14]. Data were collected from pediatric patients undergoing hemodynamic studies between April/2020 and July/2022 at the cardiac interventional area in a tertiary referral hospital in Latin America. The pediatric cardiologists working in our hospital are experienced specialists, two senior interventional cardiologists with 25 and 18 years of experience, respectively. In addition, this center participated in a multicenter study as part of the Congenital Cardiovascular Interventional Study Consortium (CCISC) [11].

The equipment used was an Artis FLOOR (Siemens Healthcare GmbH, Erlangen, Germany) installed in 2020 and equipped with a zen30 HDR detector and a high-resolution crystalline silicon matrix with 160-μm pixel size and 16-bit digitization depth, in addition to a GIGALIX X-ray tube (40–125 kV). The equipment has three acquisition protocols for pediatric cardiology examinations selected according to patient weight (i.e., CARD PED < 40 kg; CARD PED < 20 kg; and CARD PED < 6 kg). Furthermore, different cine mode settings (Im Single; 7.5 I/s; 10 I/s; 15 I/s; and 30 I/s) and various fluoroscopy mode settings (0.5–30 p/s) are available. The field sizes used during the procedures are 32 and 42 cm, respectively.

All ionizing and non-ionizing radiation-emitting equipment is supervised by the Radiological Protection Officer, who ensures compliance with the quality control program for interventional fluoroscopy equipment. Quality control is performed annually by the Radiological Protection Officer or a Medical Physicist in charge (or when equipment modifications are carried out, such as X-ray tube replacement). All tests applied to the equipment are described in detail in the IAEA-TECDOC-1958 document, which includes evaluation of environmental conditions and radiometric survey of the room, accuracy, and repeatability of the X-ray tube voltage, verification of the air kerma rate at the interventional reference point, and validation of the dose deployed by the system, as well as image quality tests [15].

The OPRIPALC methodology and latest ICRP recommendations to collect patient dose data and calculate local DRLs were used [8,10]. The first local DRLs have been obtained as the 3rd quartile values from the database containing all the collected data, as one of the options suggested by the ICRP [8]. Patient demographic data and the baseline diagnoses for which they were undergoing IPC procedures were collected. Moreover, data were collected in terms of air kerma-area product (*P*_ka_), cumulative dose magnitude (*K*_a,r_) [16,17], fluoroscopy time (FT), and several cine frames. The *P*_ka_ and *K*_a,r_ for each procedure were corrected with a calibration and mean attenuation factor of 0.85, derived from the table and mattress attenuation measured for the X-ray beam qualities used in this system for pediatric procedures [7]. The data were divided into three groups according to the type of procedure (non-complex diagnostic, complex diagnostic, and therapeutic), grouped into five age ranges (<1 year, 1 to <5 years, 5 to <10 years, 10 to <16 years, and >16 years) and five weight groups (<5 kg, 5–15 kg, 15–30 kg, 30–50 kg, and 50–80 kg) according to ICRP. According to our institution, the non-complex diagnostic procedures are generally coronography studies and some only with pressure measurement. On the contrary, complex diagnostic procedures are coronographic studies where some anatomic pathology is also sought.

The Mann–Whitney test (95% CI) was used to compare the *P*_ka_ medians for the two procedure groups (diagnostic and therapeutic). This nonparametric comparison procedure tests hypotheses and is used to find differences between two independent samples that do not necessarily have a normal distribution. Values of *p* < 0.05 were considered statistically significant [18] and STATA 16^®^ software was used [19].

## 3. Results

During the evaluation period, 255 pediatric procedures were performed (38.4% diagnostic and 61.6% therapeutic). Table 1 shows the anthropometric characteristics of the patients. 

Table 2 and Table 3 summarize the median and 3rd quartile values for *P*_ka_, *K*_a,r_, and FT magnitude for all pediatric procedures (diagnostic and therapeutic) by weight and age bands, respectively.

Table 4 shows the median *P*_ka_ values by age group reported in this work compared to that published by other groups in similar studies. 

Figure 1 and Figure 2 show a summary of 3rd quartile *P*_ka_ values (proposed as local DRLs) separated by type of procedure (diagnostic and therapeutic) and for all procedures grouped by weight and age bands, respectively. The Mann–Whitney test was used.

Table 5 groups the procedures according to type: diagnostic (complex or non-complex) and therapeutic (atrial septal defect closure, ventricular septal defect closure, patent ductus arteriosus closure, aortopulmonary collateral embolization, aortic coarctation angioplasty, aortic or pulmonary valvotomy, and others).

## 4. Discussion

Pediatric cardiac catheterization is a tool of increasing importance in the diagnosis, treatment, and follow-up of patients with congenital heart disease. However, the use of radiation has inherent procedural risks and adverse side effects that could be avoided by considering controlled radiation exposure. 

European DRL guidelines for pediatric imaging [26] suggest that generic DRL levels for diagnostic and therapeutic procedures are generally inadequate. Particularly in therapeutic procedures, higher radiation levels have been evidenced with greater variations in each procedure. Therefore, current guideline recommendations propose to create specific DRLs for pediatric interventions. In this regard, efforts have also been made in Europe to propose DRLs for IPC, highlighted in several research [20,21,23,24,27,28,29,30].

The ongoing OPRIPALC program, together with WHO, PAHO, and IAEA, a study being conducted with several reference medical centers in Latin America and the Caribbean, have evidenced that the difference in DRLs between diagnostic and therapeutic procedures was statistically significant between the age groups of children under 1 year and patients between 10 and 16 years of age [10]. Hopefully, the data and results of this initiative can be used as DRLs in different health institutions, and if levels above the established regional values are found, they can be corrected and set up in countries without defined DRLs.

In this study, a 2-year follow-up was performed in the pediatric cardiac catheterization laboratory to obtain DRLs, optimize radiation protocols, and provide local DRL values for IPC procedures in the country. All our age groups met the minimum number of 30 patients per group recommended for a DRL study [8].

The results of this paper show that radiation doses have a wide range, as expected (Table 2 and Table 3). Local DRLs for weight bands were 3.82 Gy·cm^2^ (<5 kg), 7.39 Gy·cm^2^ (5–<15 kg), 19.72 Gy·cm^2^ (15–<30 kg), 28.99 Gy·cm^2^ (30–<50 kg), and 81.71 Gy·cm^2^ (50–<80 kg), respectively. For age bands, the DRLs were 3.97 Gy·cm^2^ (<1 y), 9.94 Gy·cm^2^ (1–<5 y), 20.82 Gy·cm^2^ (5–<10 y), 58.00 Gy·cm^2^ (10–<16 y), and 31.56 Gy·cm^2^ (<16 y), respectively. In addition, it is worth noting that all *K*_a,r_ values are below 2000 mGy, which complies with the current recommendations and reduces the risk of skin lesions [31]. 

In Table 4, the comparison of *P*_ka_ median values by age range reported in this paper with others reported in similar surveys shows that the existing differences can be explained in several ways, such as in terms of the technology used, level of staff training, or procedure optimization. An example of the latter are the results achieved by Calvo Mackenna Hospital that has been involved in several IAEA programs to optimize radiation dose management in IPC procedures since 2009. An optimization program has been applied for 8 to 10 years in this hospital, which has allowed it to maintain lower dose levels than those usually published elsewhere [9]. Likewise, we consider that the results of this study allow us to continue with the process of optimizing radiological protection and lead to continuous improvement in our institution. In this regard, in order to catalyze this process of continuous improvement, it would be advisable to implement the actions described in the document “Steps for Radiation Safety in Pediatric Interventional Radiology” of Image Gently Alliance [32].

A special analysis aims to compare our current results with those reported in a previous multicenter work by Kobayashi et al. [11], in which our institution had participated. When analyzing our current DRLs values categorized by age bands (see Table 3), these were from 3.97 Gy·cm^2^ to 31.54 Gy·cm^2^, and in the previous study they were from 6.11 Gy·cm^2^ to 115.46 Gy·cm^2^.

Now, according to Figure 1 and Figure 2, statistically significant difference is shown in all weight and age bands, except for the age group between 1 and 5 years, when comparing the *P*_ka_ between diagnostic and therapeutic procedures. Also, the 3rd quartile values are proposed as local DRLs obtained in the IPC procedures by weight and age bands. 

Likewise, according to Table 5, higher *P*_ka_ and *K*_a,r_ values were found in the group of patients who underwent therapeutic procedures compared to those who underwent complex or non-complex diagnostic procedures. This could be related to the fact that during therapeutic procedures the complexity of the cases implies a longer time and higher radiation exposure dose [33]. Note that ventricular septal defect closure procedure showed the highest mean values for FT (30.3 min) and *P*_ka_ value of 37.9 Gy·cm^2^.

It is important to take into account that the process to set and update DRLs should be both flexible and dynamic. Flexibility is necessary for procedures where few data are available, as in interventional procedures in pediatric patients. A dynamic process is necessary to allow initial DRLs to be derived from these data while waiting for a wider survey to be conducted [8].

A limitation of this study is that it is a single-center study wherein, based on statistics, it was possible to establish a proposal of DRLs for diagnostic and therapeutic procedures and form classification categories based on age and weight bands, despite having a low number of procedures. Another factor to consider is the manual collection of data, which may have added some error while typing the information. To minimize the possibility of typing error, we validated the data by looking at the DICOM dose structured report for each procedure.

In pursuit of providing state-of-the-art safety and quality, our institution has updated the fluoroscopic systems used in cardiac catheterization laboratories. Integration of bi-plane systems offers superior imaging capabilities, such as acquiring two simultaneous images from two orthogonal planes and allows two cineangiography runs to be recorded simultaneously with a single injection of contrast, which should allow, among other things, to shorten procedures [34]. 

Efficient and accurate radiation data management is crucial to assess and optimize radiological protection measures. To this end, our institution has implemented an automated tool to manage radiation data. This tool is designed to gather and analyze radiation dosage data in real-time, allowing medical staff to make informed decisions during procedures, further ensuring the safety of pediatric patients.

A cornerstone of our institution’s radiological protection strategy is an ongoing educational program designed to equip healthcare professionals with the latest techniques in safeguarding children from ionizing radiation. This continuous education initiative covers various aspects, including radiation dose reduction techniques, proper use of shielding devices, and appropriate positioning during procedures.

Recognizing the varying sizes and anatomical considerations in pediatric patients, our institution places special attention on tailoring radiological protection measures that will cater to each age group. By refining the *P*_ka_/weight ratio, we account for significant differences in children’s sizes, ensuring that radiation exposure is kept to a minimum, while maintaining the required diagnostic quality. Future research will entail complementing *P*_ka_/weight ratio and refining these values for the age bands, accounting for large differences in children’s sizes. Also, proposing the Radiation Risk Categories per procedures and comparing the results with those proposed by Quinn et al. [35] can be interesting.

## 5. Conclusions

Within the framework of an international initiative (OPRIPALC) supported by WHO, PAHO, and IAEA, we obtained an initial set of institutional values of DRLs in IPC procedures for diagnostic and therapeutic procedures by weight and age groups. According to our results, radiation doses are within the ranges of other international initiatives.

It should be noted that strict radiological protection protocols have been implemented in our institution for several years, as well as high-sensitivity angiographic systems that allow imaging at low doses are used.

By establishing a baseline of radiation data for cardiac procedures and meticulously implementing comprehensive quality control procedures, the institution ensures that its interventional fluoroscopy equipment operates at the highest level of performance and safety. Regular assessments and validations of equipment functionality not only enhance the accuracy and reliability of the imaging process but also contribute to the overall radiological protection of patients, particularly children undergoing cardiac catheterization procedures.

In conclusion, our institution is committed to continuously enhancing radiological protection for children undergoing cardiac catheterization procedures. By implementing advanced fluoroscopic technology, a comprehensive education program, and tailored protection measures, we ensure that pediatric patients receive the highest standard of care, while minimizing their exposure to ionizing radiation. Through prioritizing research, collaboration, and data analysis, we strive to set new benchmarks for radiological protection in pediatric cardiology.

To our knowledge, this is one of the first proposals for establishing local DRLs in PIC procedures in Colombia, and we hope that it will serve as a starting point to continue with the efforts to have DRLs at a national level, as well as to implement optimization actions in these procedures in pediatrics.

## Figures and Tables

**Figure 1 children-10-01588-f001:**
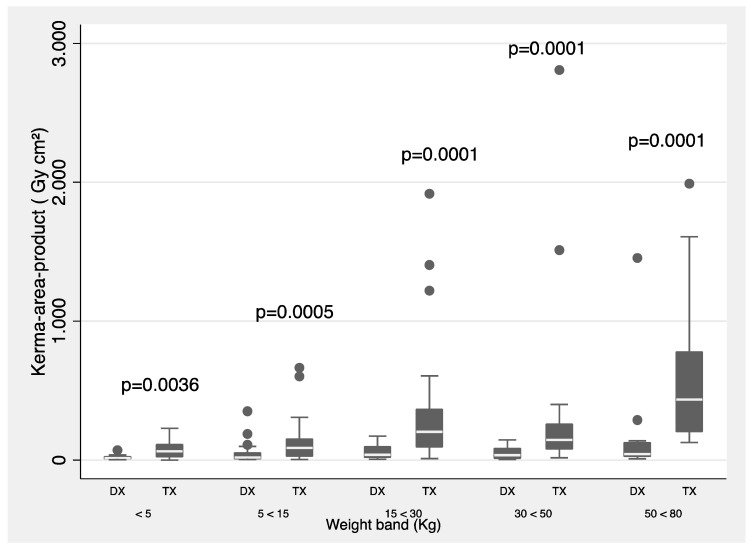
*P*_ka_ values for diagnostic (DX) and therapeutic (TX) procedures. The data are for the <5 kg, 5–<15 kg, 15–<30 kg, 30–<50 kg, and 50–<80 kg weight bands. Boxes represent the interquartile range; the vertical bars extend to the highest and lowest values.

**Figure 2 children-10-01588-f002:**
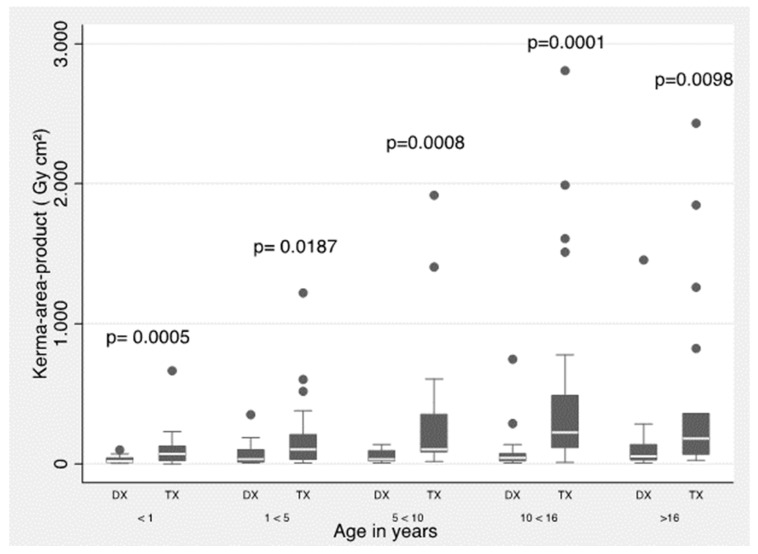
*P*_ka_ values for diagnostic (DX) and therapeutic (TX) procedures. The data are for bands <1 year, 1–<5 years, 5–<10 years, 10–<16 years, and >16 years. Boxes represent the interquartile range; the vertical bars extend to the highest and lowest values.

**Table 1 children-10-01588-t001:** Sample size (*n*), median values of age, height, and weight by weight bands.

Weight Band (kg)	N	Age in Years (yr)	Height (cm)	Weight (kg)
<5	43	0.63	50.37	3.33
5–<15	78	2.21	75.09	9.14
15–<30	59	6.05	112.40	20.01
30–<50	41	13.07	146.95	39.71
50–<80	34	17.72	161.13	57.99

**Table 2 children-10-01588-t002:** Sample size (*n*), median and 3rd quartile values for kerma-area product (*P*_ka_), cumulative air kerma at patient entrance reference point (*K*_a,r_), and fluoroscopy time (FT) by weight bands.

Weight Band (kg)	*n*	*P*_ka_ (Gy·cm^2^) Median–3rd	*K*_a,r_ (mGy) Median–3rd	FT (min) Median–3rd
<5	43	1.69–3.82	37.00–86.88	13.27–19.73
5–<15	78	3.44–7.39	50.00–119.13	8.42–16.24
15–<30	41	8.87–19.72	95.80–211.0	7.00–18.01
30–<50	59	10.51–28.99	100.65–199.57	8.00–17.10
50–<80	34	22.61–81.71	171.00–491.35	12.05–21.58

**Table 3 children-10-01588-t003:** Sample size (*n*), median and 3rd quartile values for kerma-area product (*P*_ka_), cumulative air kerma at patient entrance reference point (*K*_a,r_), and fluoroscopy time (FT) by age bands.

Age Band (yr)	*n*	*P*_ka_ (Gy·cm^2^) Median–3rd	*K*_a,r_ (mGy) Median–3rd	FT (min) Median–3rd
<1	66	1.96–3.97	43.65–101.50	10.68–17.90
1–<5	68	4.67–9.94	73.30–133.65	8.37–15.36
5–<10	45	8.16–20.82	92.95–204.13	10.68–18.80
10–<16	46	14.76–58.00	124.30–319.20	11.41–23.77
<16	30	17.52–31.56	103.80–205.35	8.29–14.27

**Table 4 children-10-01588-t004:** Comparison of kerma-area product (*P*_ka_) and median values (Gy cm^2^) for IPC procedures reported in this paper and others (values adapted by the authors of this paper) *.

Age Group (years)	Martinez et al., 2007 [20]	Verghese et al., 2012 [21]	Ubeda et al., 2012 [9]	Corredoira et al., 2015 [22]	Kottou et al., 2018 [23]	Ubeda et al., 2020 [24]	Ishibashi et al., 2021 [25]	Ubeda et al., 2022 [10]	This Paper, 2023
<1	1.9	4.6	0.9	1.8	2.0	2.1	4.3	1.9	1.8
1–<5	2.9	8.3	1.5	3.1	3.0	4.7	6.3	2.6	4.5
5–<10	4.5	11.5	2.1	6.0	7.0	6.3	10.9	3.6	7.5
10–<16	15.4	24.7	5.0	12.1	14.0	13.6	19.4	11.5	13.4

* Note that the comparison is carried out for median values and not for DRLs (3rd quartile) because the 3rd quartile values are not reported in all the papers.

**Table 5 children-10-01588-t005:** Frequencies, median (1st–3rd quartiles) values for kerma-area product (*P*_ka_) and fluoroscopy time (FT) for each type of procedure performed.

Procedure Type	Procedure Name	Frequency	*P*_ka_ (Gy·cm^2^) Median (1st–3rd)	FT (min) Median (1st–3rd)
Diagnostic	Non-complex	61	2.0 (1.2–5.5)	5.1 (3.5–9.4)
	Complex	37	4.5 (1.12–11.7)	7.2 (5.2–14.4)
Therapeutic	Atrial septal defect closure	12	9.0 (3.7–10.8)	6.7 (5.2–10.2)
	Ventricular septal defect closure	9	37.9 (6.1–74.2)	30.3 (17.0–58.5)
	Patent ductus arteriosus closure	51	7.3 (2.9–13.5)	7.5 (5.4–11.9)
	Aortopulmonary collateral embolization	22	8.6 (1.9–31.3)	17.7 (10.2–26.8)
	Aortic coarctation angioplasty	9	8.0 (0.7–25.4)	12.8 (33.0–58.0)
	Aortic or pulmonary valvuloplasty	33	9.7 (2.1–39.1)	17.6 (9.5–26.4)
	Other	21	3.9 (0.8–33.1)	15.4 (11.3–30.6)

## Data Availability

Not applicable.

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
