# Peer review of "Diagnostic Reference Levels in Interventional Pediatric Cardiology: Two-Year Experience in a Tertiary Referral Hospital in Latin America"

_children, 2023, doi:10.3390/children10101588_

Round 1

Reviewer 1 Report

Your work is interesting and I think it will prove to be a real asset in your field. It is very important in our field of work to set limits to ensure the wellbeing of our patients and to have protocols.

But for your work to be taken into consideration for publication in such a prestigious journal you need to make some serious efforts to make it original.

You need to rephrase the introduction, discussion and conclusion sections because of high percentage of plagiarism detected.

Abstract

-         Row 24 – perhaps without the word “data” just the 3rd quartile of patient dose distribution for…;

-         Row 26 – perhaps it would sound better if you choose another synonym for “bands”;

Introduction

-         Row 49 – what is the objective? related to the phrase before it seems like there are two “ avoid deterministic effects and reduce stochastic effects” but the following phrase talks about reducing one single objective

-         Row 64 – perhaps “for” instead of “of” will be better

-         Row 52 - why there are no dose limits applied to patients?

-         Row 62 – add please local DRL or typical value – to enhance understanding

Materials and methods

-         Rows 86-87 rephrase – the study was an observational descriptive case series and you need informed consent or it was a retrospective one with all the data gathered retrospectively and you don’t need informed consent; its either one or the other;

-         Row 95 – if the equipment has 3 acquisition protocols depending on one’s weight why do you later decided to have 5 weight groups? The different types of acquisition protocols do not influence your results? Perhaps you should analyze also the differences between the three protocols of acquisition

-         Rows 97-98 make it a sentence – there is no verb in that sentence

-         Row 105 – include with an “s” – includes

-         Rows 12-127 - delete the phrase, it is unnecessary

-         Rows 129-131 – rephrase according to the prior suggested modification; decide either prospective either retrospective

Results

-         Table 1 – delete the highlight in yellow and delete the separation between rows according to the journal specifications; do not include table title in the actual table it should be above the table

-         Same comments for all the tables – move the title above the actual table, outside the table and delete the lines between the rows of the table

-         Table 4 – move the supplementary information and notes under the table not in its title; also change the reference numbers from the table to the first authors name followed by et al and the year of publication – ex Martinez et al.,  2007

The references need to respect the journal specifications, they are not written accordingly.

There are some very difficult-to-understand phrases.

Author Response

Your work is interesting and I think it will prove to be a real asset in your field. It is very important in our field of work to set limits to ensure the wellbeing of our patients and to have protocols.

Corresponding author answer:

Thank you for the positive opinion regarding this paper.

But for your work to be taken into consideration for publication in such a prestigious journal you need to make some serious efforts to make it original.

You need to rephrase the introduction, discussion and conclusion sections because of high percentage of plagiarism detected.

Corresponding author answer:

Thanks for the comment. We have rephrased the following texts, for the introduction and discussion sections:

Introduction

The development of cardiology has led to increasing numbers of interventional pro-cedures year after year. Over the past few decades, interventional pediatric cardiology (IPC) has transitioned from primarily a diagnostic instrument to a therapeutic approach that has markedly enhanced the prognosis for congenital heart defects [1,2].

Therefore, radiation protection strategies exist to avoid deterministic effects and reduce, as much as possible, the probability of stochastic effects [3,5]. These strategies is embodied in three principles, that are part of the radiation protection system: (1) justification of the study by being clear that the exposure produces a net benefit versus the potential risks; (2) limitation of the dose to avoid exceeding established values (only applicable to personnel and the general public, no dose limits are applied to patients); and (3) optimization to achieve doses as low as reasonably achievable, called the "ALARA" concept [5,6].

This study was carried out within the framework of the initiative "Optimization of Protection in Pediatric Interventional Radiology in Latin America and the Caribbean" (OPRIPALC), instituted in 2018. This program is a collaborative endeavor by the Pan-American Health Organization (PAHO) and the World Health Organization (WHO), in association with the International Atomic Energy Agency (IAEA), to assist their mem-ber countries in guaranteeing that the radiation exposure of pediatric patients is kept to a bare minimum during fluoroscopy-aided interventional procedures [13,14].

Discussion

The results of this paper show that radiation doses have a wide range, as expected (Tables 2 and 3). Local DRLs were 3.82 Gy·cm2 (< 5 kg), 7.39 Gy·cm2 (5 - < 15 kg), 19.72 Gy·cm2 (15 - < 30 kg), 28.99 Gy·cm2 (30 - < 50 kg), and 81.71 Gy·cm2 (50-<80 kg), respectively. For age bands, the DRLs were 3.97 Gy·cm2 (< 1 y), 9.94 Gy·cm2 (1 - < 5 y), 20.82 Gy·cm2 (5 - < 10 y), 58.00 Gy·cm2 (10 - < 16 y), and 31.56 Gy·cm2 (< 16 y), respectively. In addition, it is worth noting that all Ka,r values are below 2,000 mGy, which complies with current recommendations and reduces the risk of skin lesions [28].

In this sense, in order to catalyze this process of continuous improvement, it would be advisable to implement the actions described in the document "Steps for Radiation Safety in Pediatric Interventional Radiology" of Image Gently Alliance [29].

A special analysis aims to compare our current results with those reported in a previous multicenter work by Kobayashi et al., [12] in which our institution had participated. When analyzing our current DRLs values categorized by age bands (see Table 3), these were from 3.97 Gy·cm2 to 31.54 Gy·cm2 and in previous study it were from 6.11 Gy·cm2 to 115.46 Gy·cm2, respectively.

In pursuit of providing state-of-the-art safety and quality, our institution has updated the fluoroscopic systems used in cardiac catheterization laboratories. Integration of bi-plane systems offers superior imaging capabilities, such as: acquiring two simultaneous images from two orthogonal planes and imaging allows two cineangiography runs to be recorded simultaneously with a single injection of contrast, which should allow among other things shorten procedures [31].

Future work includes complementing with Pka/Weight ratio and refining these values for the age bands, accounting for large differences in children’s sizes. Also, proposing Radiation Risk Categories per procedures and compare your results with those proposed by Quinn et al., [32] can be interesting.

Abstract

-         Row 24 – perhaps without the word “data” just the 3rd quartile of patient dose distribution for…;

Corresponding author answer:

Thanks for the comment. Corrected.

-         Row 26 – perhaps it would sound better if you choose another synonym for “bands”;

Corresponding author answer:

Thanks for the comment. However, we prefer to continue using the terminology proposed by the International Commission on Radiological Protection in its report 135.

Introduction

-         Row 49 – what is the objective? related to the phrase before it seems like there are two “ avoid deterministic effects and reduce stochastic effects” but the following phrase talks about reducing one single objective

Corresponding author answer:

Thanks for the comment. We have added the following new text to the introduction section:

Therefore, radiation protection strategies exist to avoid deterministic effects and reduce, as much as possible, the probability of stochastic effects [3,5]. These strategies is embodied in three principles, that are part of the radiation protection system: (1) justification of the study by being clear that the exposure produces a net benefit versus the potential risks; (2) limitation of the dose to avoid exceeding established values (only applicable to personnel and the general public, no dose limits are applied to patients); and (3) optimization to achieve doses as low as reasonably achievable, called the "ALARA" concept [5,6].

-         Row 64 – perhaps “for” instead of “of” will be better

Corresponding author answer:

Thanks for the comment. Corrected.

-         Row 52 - why there are no dose limits applied to patients?

Corresponding author answer:

Thanks for the comment. Because it is assumed that the Physicians justified the detriment that exposure to radiation could implies for the patient. Likewise, limiting the amount of radiation used during a radiology or interventional procedure can cause a loss in the quality of diagnostic or therapeutic information of the images produced.

For more detail you can review the report 103 of the International Commission on Radiological Protection (7. MEDICAL EXPOSURE OF PATIENTS, COMFORTERS AND CARERS, AND VOLUNTEERS IN BIOMEDICAL RESEARCH).

-         Row 62 – add please local DRL or typical value – to enhance understanding

Corresponding author answer:

Thanks for the comment. Added

Materials and methods

-         Rows 86-87 rephrase – the study was an observational descriptive case series and you need informed consent or it was a retrospective one with all the data gathered retrospectively and you don’t need informed consent; its either one or the other;

Corresponding author answer:

Thanks for the comment. We have corrected and added the following text.

The observational study design was retrospective case series [15]. Data was collected from pediatric patients undergoing hemodynamic studies between April/2020 and July/2022 at the cardiac interventional area in a tertiary referral hospital in Latin America.

-         Row 95 – if the equipment has 3 acquisition protocols depending on one’s weight why do you later decided to have 5 weight groups? The different types of acquisition protocols do not influence your results? Perhaps you should analyze also the differences between the three protocols of acquisition

Corresponding author answer:

Thanks for the comment. Because the recommendation of the International Commission on Radiological Protection (report 135) is to carry out the analysis by age group and weight, not by each type of protocol that is specific to each center. The establishment of the DRLs must be by weight and/or age bands, not by the protocol used. Another different thing would be to evaluate each protocol in terms of dose and image quality, but it is not within the objectives of this work.

-         Rows 97-98 make it a sentence – there is no verb in that sentence

Corresponding author answer:

Thanks for the comment. We have added the following text.

The equipment used was an Artis FLOOR (Siemens Healthcare GmbH) installed in 2020 and equipped with a zen30 HDR detector with a high-resolution crystalline silicon matrix with 160-μm pixel size and 16-bit digitization depth, in addition to a GIGALIX X-ray tube (40 – 125 kV). The equipment has three acquisition protocols for pediatric cardiology examinations selected according to patient weight (i.e., CARD PED < 40 kg; CARD PED < 20 kg, and CARD PED < 6 kg). Furthermore, different cine mode settings (Im Single; 7.5I/s; 10I/s; 15I/s, and 30I/s) and various fluoroscopy mode settings (0.5 - 30p/s) are available. The field sizes used during the procedures are 32 and 42 cm, respectively.

-         Row 105 – include with an “s” – includes

Corresponding author answer:

Thanks for the comment. Added

-         Rows 12-127 - delete the phrase, it is unnecessary

Corresponding author answer:

Thanks for the comment. However, I don't understand what the starting line is. If you like, I ask you to please delete the text directly to speed up the process of improving the article.

-         Rows 129-131 – rephrase according to the prior suggested modification; decide either prospective either retrospective

Corresponding author answer:

Thanks for the comment. Following your advice, we have decided to completely remove that text.

Results

-         Table 1 – delete the highlight in yellow and delete the separation between rows according to the journal specifications; do not include table title in the actual table it should be above the table

Corresponding author answer:

Thanks for the comment. We have corrected.

-         Same comments for all the tables – move the title above the actual table, outside the table and delete the lines between the rows of the table

Corresponding author answer:

Thanks for the comment. We have corrected and deleted the line between the rows of the tables.

Table 1. Sample size (n), median values of age, height, and weight by weight band.

Weight band (kg)

N

Age in years (yr)

Height (cm)

Weight (kg)

< 5

43

0.63

50.37

3.33

5 - < 15

78

2.21

75.09

9.14

15 - < 30

59

6.05

112.40

20.01

30 - < 50

41

13.07

146.95

39.71

50 - < 80

34

17.72

161.13

57.99

Table 2. Sample size (n), median and 3rd quartile values for kerma-area product (Pka), cumulative air kerma at patient entrance reference point (Ka,r), and fluoroscopy time (FT) by weight band.

Weight band (kg)

Pka (Gy·cm2

Median – 3rd

Ka,r (mGy)

Median – 3rd

FT (min)

Median – 3rd

< 5 

43 

1.69 - 3.82 

37.00 - 86.88 

13.27 - 19.73 

5 - < 15 

78 

3.44 - 7.39 

50.00 - 119.13 

8.42 - 16.24 

15 - < 30 

41 

8.87 - 19.72 

95.80 - 211.0 

7.00 - 18.01 

30 - < 50 

59 

10.51 - 28.99 

100.65 - 199.57 

8.00 - 17.10 

50 - < 80

34 

22.61 - 81.71 

171.00 - 491.35 

12.05 - 21.58 

Table 3. Sample size (n), median and 3rd quartile values for kerma-area product (Pka), cumulative air kerma at patient entrance reference point (Ka,r), and fluoroscopy time (FT) by age band.

Age band (yr)

Pka (Gy·cm2

Median - 3rd 

Ka,r (mGy)

Median - 3rd 

FT (min)

Median - 3rd 

< 1

66

1.96 – 3.97

43.65 – 101.50

10.68 – 17.90

1 - < 5 

68

4.67 – 9.94

73.30 – 133.65

8.37 – 15.36

5 - < 10 

45

8.16 – 20.82

92.95 – 204.13

10.68 – 18.80

10 - < 16 

46 

14.76 – 58.00

124.30 – 319.20

11.41 – 23.77

< 16

30 

17.52 – 31.56

103.80 – 205.35

8.29 – 14.27

Table 4. Comparison of kerma-area product (Pka) median values (Gy•cm2) for IPC procedures reported in this and other papers (values adapted by the authors of this paper). Note that the comparison is made for median values and not for DRLs (3rd quartile) because the 3rd quartile values are not reported in all the papers.

Age group (years)

Martinez et al., 2007

Verghese et al., 2012

Ubeda et al., 2012

Corredoira et al., 2015

Ubeda et al. , 2018

Kottou et al., 2018

Ubeda et al., 2020

Ishibashi et al. , 2021

Ubeda et al., 2022

This paper, 2023

< 1

1.9

4.6

0.9

1.8

1.1

2.0

2.1

4.3

1.9

1.8

1 - < 5

2.9

8.3

1.5

3.1

1.5

3.0

4.7

6.3

2.6

4.5

5 - < 10

4.5

11.5

2.1

6.0

2.7

7.0

6.3

10.9

3.6

7.5

10 - < 16

15.4

24.7

5.0

12.1

8.4

14.0

13.6

19.4

11.5

13.4

Table 5. Frequencies, median (1rd - 3rd quartiles) values for kerma-area product (Pka) and fluoroscopy time (FT) for each type of procedure performed.

Procedure type

Procedure name

Frequency

Pka (Gy·cm2) Median (1rd - 3rd)

FT (min) Median (1rd - 3rd)

Diagnostic

Non-complex

61

2.0(1.2-5.5)

5.1(3.5-9.4)

Complex

37

4.5(1.12-11.7)

7.2(5.2-14.4)

Therapeutic

Atrial septal defect closure

12

9.0(3.7-10.8)

6.7(5.2-10.2)

Ventricular septal defect closure

9

37.9(6.1-74.2)

30.3(17.0-58.5)

Patent ductus arteriosus closure

51

7.3(2.9-13.5)

7.5(5.4-11.9)

Aortopulmonary collateral embolization

22

8.6(1.9-31.3)

17.7(10.2-26.8)

Aortic coarctation angioplasty

9

8.0(0.7-25.4)

12.8(33.0-58.0)

Aortic or pulmonary valvuloplasty

33

9.7(2.1-39.1)

17.6(9.5-26.4)

Other

21

3.9(0.8-33.1)

15.4(11.3-30.6)

-         Table 4 – move the supplementary information and notes under the table not in its title; also change the reference numbers from the table to the first authors name followed by et al and the year of publication – ex Martinez et al.,  2007

Corresponding author answer:

Thanks for the comment. We have corrected and added the first authors name followed by the year of publication and erased the reference numbers.

Table 4. Comparison of kerma-area product (Pka) median values (Gy•cm2) for IPC procedures reported in this and other papers (values adapted by the authors of this paper). Note that the comparison is made for median values and not for DRLs (3rd quartile) because the 3rd quartile values are not reported in all the papers.

Age group (years)

Martinez et al., 2007

Verghese et al., 2012

Ubeda et al., 2012

Corredoira et al., 2015

Ubeda et al. , 2018

Kottou et al., 2018

Ubeda et al., 2020

Ishibashi et al. , 2021

Ubeda et al., 2022

This paper, 2023

< 1

1.9

4.6

0.9

1.8

1.1

2.0

2.1

4.3

1.9

1.8

1 - < 5

2.9

8.3

1.5

3.1

1.5

3.0

4.7

6.3

2.6

4.5

5 - < 10

4.5

11.5

2.1

6.0

2.7

7.0

6.3

10.9

3.6

7.5

10 - < 16

15.4

24.7

5.0

12.1

8.4

14.0

13.6

19.4

11.5

13.4

Reviewer 2 Report

Aristizabal et al have done a very good job with this paper and should be commended on this worthwhile effort.  This project is a local and national effort in Colombia to describe and thereby establish acceptable radiation doses for pediatric cardiac catheterizations through an observational, descriptive study with retrospective collection and analysis of data.  There is a considerable number of procedures collected in this study (255 procedures).  The data are clearly presented and the methods are adequately described.  As the authors acknowledge, there is a paucity of data on radiation safety for pediatric patients undergoing cardiac catheterization, particularly in centers/regions outside of the U.S. and Europe, which makes the novelty and merit of this work high, in my opinion while acknowledging that this study is a descriptive retrospective analysis.  Overall, I believe this manuscript should be considered for publication with minor revisions as outlined below. 

1.  The first two paragraphs are broad and a bit vague. I would recommend condensing these paragraphs and focus only on the use of radiation in interventional pediatric cardiology and the potential risks. The statement “IPC procedures can lead to a high risk of developing cancer due to their longer life expectancy” is particularly unclear and misleading. The actual risk is unknown for patients with congenital heart disease.  The phrase “their longer life expectancy” is not clear to whom it is referring and confusing to state a higher risk of cancer concomitantly with a longer life expectancy... I would remove this statement.

2.  In the Discussion section, the experience of this center in the CCISC is mentioned and reference cited. I would suggest that this experience be mentioned and cited in the Methods section. The use of air kerma product (PKA) in this study, is in part, due to the published findings of the CCISC as well as other multi-center studies. However, related to that publication, I believe that this study’s analysis may be strengthened if a descriptive analysis were performed for PKA by body weight.  The analysis by age bands and by procedure type is clinically useful, but I feel the analysis by weight is less useful. 

3.  I believe that one of the areas that is most clinically useful and represents future opportunities for quality improvement is the analysis by case type.  How are diagnostic cases determined to be complex? I don’t believe this is described but should be mentioned in the Methods. However, is making this distinction needed?  Analyses performed do not seem to use this sub-grouping and perhaps it would be clearer to just have a category for all diagnostic cases. 

Overall on this point, I would recommend that the authors consider using some established data for procedure-specific radiation risk which I believe would add significant utility to the paper.  Here is a reference to consider:

Quinn BP, Armstrong AK, Bauser-Heaton HD, Callahan R, El-Said HG, Foerster SR, Goldstein BH, Goodman AS, Gudausky TM, Kreutzer JN, Leahy RA, Petit CJ, Rockefeller TA, Shahanavaz S, Trucco SM, Bergersen L; Congenital Cardiac Catheterization Project on Outcomes-Quality Improvement (C3PO-QI). Radiation Risk Categories in Cardiac Catheterization for Congenital Heart Disease: A Tool to Aid in the Evaluation of Radiation Outcomes. Pediatr Cardiol. 2019 Feb;40(2):445-453. doi: 10.1007/s00246-018-2024-3. Epub 2018 Nov 30. PMID: 30506273.

I think it would be very worthwhile to consider an analysis based on this multi-center data from the U.S., and to mention the risk category of the procedures performed in Colombia.  An analysis would need to include comparable units (DAP/kg), which connects to #2.  It is possible that the data from Colombia compares quite favorably to radiation doses at U.S. centers, and I believe that would be a noteworthy finding. 

4.  In the Conclusions section, it is mentioned that biplane systems offer reduction in radiation.  Is there evidence to support that statement?  A lateral camera often contributes to a higher dose of radiation.  Perhaps this paragraph of the discussion is focused on the technological advancement of fluoroscopic systems, which have effectively lowered radiation doses while maintaining image quality.

5.  Table 5 needs to include IQR with the median values.  Please capitalize all procedure names for consistency. 

Line 44-45:  Delete “; this: at the end of line 44. 

Line 125: Not necessary to describe the Mann-Whitney test.  Would remove this sentence.

Line 197:  Paragraph starting on line 197 is not very clearly written.  Would consider re-writing this paragraph.  

Line 204:  “weight and age bands”.

Line 269: Change PIC to IPC.

Author Response

Aristizabal et al have done a very good job with this paper and should be commended on this worthwhile effort.  This project is a local and national effort in Colombia to describe and thereby establish acceptable radiation doses for pediatric cardiac catheterizations through an observational, descriptive study with retrospective collection and analysis of data.  There is a considerable number of procedures collected in this study (255 procedures).  The data are clearly presented and the methods are adequately described.  As the authors acknowledge, there is a paucity of data on radiation safety for pediatric patients undergoing cardiac catheterization, particularly in centers/regions outside of the U.S. and Europe, which makes the novelty and merit of this work high, in my opinion while acknowledging that this study is a descriptive retrospective analysis.  Overall, I believe this manuscript should be considered for publication with minor revisions as outlined below. 

 Corresponding author answer:

Thank you for the positive opinion regarding this paper.

  1. The first two paragraphs are broad and a bit vague. I would recommend condensing these paragraphs and focus only on the use of radiation in interventional pediatric cardiology and the potential risks. The statement “IPC procedures can lead to a high risk of developing cancer due to their longer life expectancy” is particularly unclear and misleading. The actual risk is unknown for patients with congenital heart disease.  The phrase “their longer life expectancy” is not clear to whom it is referring and confusing to state a higher risk of cancer concomitantly with a longer life expectancy... I would remove this statement.

Corresponding author answer:

Thanks for the comment. However, as we understand it, an introduction should go from the general to the particular, for this reason perhaps you find the first two paragraphs too broad and a little vague. Despite this response from us to your comment, please review how we have redacted these first two paragraphs again, deleting the suggested phrase.

The development of cardiology has led to increasing numbers of interventional procedures year after year. Over the past few decades, interventional pediatric cardiology (IPC) has transitioned from primarily a diagnostic instrument to a therapeutic approach that has markedly enhanced the prognosis for congenital heart defects [1,2].

However, it is known that these procedures may involve high doses of radiation, which increases the potential risk of stochastic effects, especially in pediatric patients; given the higher sensitivity of their tissues compared to adults [3, 4].

  1. In the Discussion section, the experience of this center in the CCISC is mentioned and reference cited. I would suggest that this experience be mentioned and cited in the Methods section. The use of air kerma product (PKA) in this study, is in part, due to the published findings of the CCISC as well as other multi-center studies. However, related to that publication, I believe that this study’s analysis may be strengthened if a descriptive analysis were performed for PKA by body weight.  The analysis by age bands and by procedure type is clinically useful, but I feel the analysis by weight is less useful. 

 Corresponding author answer:

Thanks for the comments. We have previously cited the experience of our centers in the introduction section. However, we have added the following new text to the Materials and Methods section:

The observational study design was retrospective case series [15]. Data was collected from pediatric patients undergoing hemodynamic studies between April/2020 and July/2022 at the cardiac interventional area in a tertiary referral hospital in Latin America. The pediatric cardiologists working in our hospital are experienced specialists, with two senior interventional cardiologists, with 25 and 18 years of experience, respectively. In addition, this center participated in a multicenter study, as part of the Congenital Cardiovascular Interventional Study Consortium (CCISC) [12].

On the other hand, we do not share your opinion that the analysis by weight variable is less important for the radiation doses, especially because in this work we are following the strict recommendations of the International Commission on Radiological Protection in its report 135.

  1. I believe that one of the areas that is most clinically useful and represents future opportunities for quality improvement is the analysis by case type.  How are diagnostic cases determined to be complex? I don’t believe this is described but should be mentioned in the Methods. However, is making this distinction needed?  Analyses performed do not seem to use this sub-grouping and perhaps it would be clearer to just have a category for all diagnostic cases. 

Corresponding author answer:

Thanks for the comment. We have added the following text in material and method section:

According to our institution, the non-complex diagnostic procedures are generally coronography studies and some only with pressure measurement. On the contrary, complex diagnostic procedures are coronographic studies where some anatomic pathology is also sought.

Overall on this point, I would recommend that the authors consider using some established data for procedure-specific radiation risk which I believe would add significant utility to the paper.  Here is a reference to consider:

Quinn BP, Armstrong AK, Bauser-Heaton HD, Callahan R, El-Said HG, Foerster SR, Goldstein BH, Goodman AS, Gudausky TM, Kreutzer JN, Leahy RA, Petit CJ, Rockefeller TA, Shahanavaz S, Trucco SM, Bergersen L; Congenital Cardiac Catheterization Project on Outcomes-Quality Improvement (C3PO-QI). Radiation Risk Categories in Cardiac Catheterization for Congenital Heart Disease: A Tool to Aid in the Evaluation of Radiation Outcomes. Pediatr Cardiol. 2019 Feb;40(2):445-453. doi: 10.1007/s00246-018-2024-3. Epub 2018 Nov 30. PMID: 30506273.

I think it would be very worthwhile to consider an analysis based on this multi-center data from the U.S., and to mention the risk category of the procedures performed in Colombia.  An analysis would need to include comparable units (DAP/kg), which connects to #2.  It is possible that the data from Colombia compares quite favorably to radiation doses at U.S. centers, and I believe that would be a noteworthy finding. 

 Corresponding author answer:

Thanks for the comments. Although your proposal is highly interesting, we believe that it is far from the central objective of this article, which is “to obtain the first local DRLs in our institution to perform a self-assessment and optimization of radiation protocols, in addition to contributing to the establishment of national DRLs for IPC”.  These DRLs are classified for weight and age bands and proposed for diagnostic and therapeutic procedures. In addition, as indicated in lines 243-245, the use of the relationship DAP/kg will be part of the next work we do in our center. In relation to the use of data for procedure-specific radiation risk, it also seems to us a very interesting idea to develop in future works. To incorporate those ideas, we have added the following texts and reference in the discussion section.

Recognizing the varying sizes and anatomical considerations in pediatric patients, our institution places special attention on tailoring radiological protection measures to each age group. By refining the Pka/Weight ratio, we account for significant differences in children’s sizes, ensuring that radiation exposure is kept to a minimum while maintaining the required diagnostic quality. Future work includes complementing with Pka/Weight ratio and refining these values for the age bands, accounting for large differences in children’s sizes. Also, proposing Radiation Risk Categories per procedures and compare your results with those proposed by Quinn et al., [32] can be interesting.

  1. Quinn, B.; Armstrong, A.; Bauser-Heaton, H.; Callahan, R.; El-Said, H.; Foerster, S.; Goldstein, B.; Goodman, A.; Gudausky, T.: Kreutzer, J.; Congenital Cardiac Catheterization Project on Outcomes-Quality Improvement (C3PO-QI) (2019). Radiation Risk Categories in Cardiac Catheterization for Congenital Heart Disease: A Tool to Aid in the Evaluation of Radiation Outcomes. Pediatric. cardiology, 40(2); 445–453. https://doi.org/10.1007/s00246-018-2024-3
  2. In the Conclusions section, it is mentioned that biplane systems offer reduction in radiation.  Is there evidence to support that statement?  A lateral camera often contributes to a higher dose of radiation.  Perhaps this paragraph of the discussion is focused on the technological advancement of fluoroscopic systems, which have effectively lowered radiation doses while maintaining image quality.

 Corresponding author answer:

Thanks for the comments. By performing a more exhaustive review of the bibliography, we found references that support what you have indicated. We have rewritten the text as follows.

In pursuit of providing state-of-the-art safety and quality, our institution has updated the fluoroscopic systems used in cardiac catheterization laboratories. Integration of bi-plane systems offers superior imaging capabilities, such as: acquiring two simultaneous images from two orthogonal planes and imaging allows two cineangiography runs to be recorded simultaneously with a single injection of contrast, which should allow among other things shorten procedures [31].

  1. Table 5 needs to include IQR with the median values.  Please capitalize all procedure names for consistency. 

Corresponding author answer:

Thanks for the comments. We have added the IQR values to table 5 and capitalized all procedure names.

Table 5. Frequencies, median (1rd - 3rd quartiles) values for kerma-area product (Pka) and fluoroscopy time (FT) for each type of procedure performed.

Procedure type

Procedure name

Frequency

Pka (Gy·cm2) Median (1rd - 3rd)

FT (min) Median (1rd - 3rd)

Diagnostic

Non-complex

61

2.0(1.2-5.5)

5.1(3.5-9.4)

Complex

37

4.5(1.12-11.7)

7.2(5.2-14.4)

Therapeutic

Atrial septal defect closure

12

9.0(3.7-10.8)

6.7(5.2-10.2)

Ventricular septal defect closure

9

37.9(6.1-74.2)

30.3(17.0-58.5)

Patent ductus arteriosus closure

51

7.3(2.9-13.5)

7.5(5.4-11.9)

Aortopulmonary collateral embolization

22

8.6(1.9-31.3)

17.7(10.2-26.8)

Aortic coarctation angioplasty

9

8.0(0.7-25.4)

12.8(33.0-58.0)

Aortic or pulmonary valvuloplasty

33

9.7(2.1-39.1)

17.6(9.5-26.4)

Other

21

3.9(0.8-33.1)

15.4(11.3-30.6)

Line 44-45:  Delete “; this: at the end of line 44. 

Corresponding author answer:

Thanks for the comments. Delete.

Line 125: Not necessary to describe the Mann-Whitney test.  Would remove this sentence.

Corresponding author answer:

Thanks for the comments. However, as far as we know, it is important to indicate the statistical test used to compare the medians, since the statistical analysis performed must be described in detail in the material and method section. We respectfully suggest that we can leave the text as is.

Line 197:  Paragraph starting on line 197 is not very clearly written.  Would consider re-writing this paragraph.  

Corresponding author answer:

Thanks for the comments. We have added the following new text:

A special analysis aims to compare our current results with those reported in a previous multicenter work by Kobayashi et al., [12] in which our institution had participated. When analyzing our current DRLs values categorized by age bands (see Table 3), these were from 3.97 Gy·cm2 to 31.54 Gy·cm2 and in previous study it were from 6.11 Gy·cm2 to 115.46 Gy·cm2, respectively.

Line 204:  “weight and age bands”.

Corresponding author answer:

Thanks for the comments. Corrected.

Line 269: Change PIC to IPC.

Corresponding author answer:

Thanks for the comments. Corrected.

Reviewer 3 Report

The article outlines diagnostic reference levels for interventional pediatric cardiology,  which will help develop relevant national priority programs for Colombia in the near future and it is very meaningful. However, some contents of the manuscript still need to be improved:

1. In this manuscript, they claimed that these results were compared with other existing DRL values and found that the results were similar to those of other centers, and there was more space for further optimization of radiation dose values. It is suggested to supplement the comparison of this part of the data in the paper and to discuss the advantages and disadvantages of this part of the data.

2. In this manuscript, the included data is only the data of one hospital in Colombia. It is suggested to combine the relevant data of other hospitals for unified analysis and comparison, and increase the sample size, which will be more convincing and provide guiding significance.

This manuscript required  minor editing of English language.

Author Response

Thanks for the comment.

  1. In this manuscript, they claimed that these results were compared with other existing DRL values and found that the results were similar to those of other centers, and there was more space for further optimization of radiation dose values. It is suggested to supplement the comparison of this part of the data in the paper and to discuss the advantages and disadvantages of this part of the data.

Corresponding author answer:

Thanks for the comment. However, we have not fully understood your comment, as we think the advantages and disadvantages of our results and comparisons with other works have been addressed in the other sections of the discussion. But we have added the following text that we think can improve the work and is in line with what you commented.

In Table 4, with the comparison of Pka median values by age range reported in this paper and others reported in similar surveys, differences exist that can be explained in several ways, from differences in terms of the technology used, level of staff training, or procedure optimization. An example of the latter, are the results achieved by Calvo Mackenna Hospital that has been involved in several IAEA programs to optimize radiation dose management in IPC procedures since 2009. An optimization program has been applied for 8 to 10 years at this hospital, which has allowed it to maintain lower dose levels than those usually published elsewhere [9]. Likewise, we consider that the results of this study allow us to continue with the process of optimization of radiological protection and continuous improvement of our institution. In this sense, in order to catalyze this process of continuous improvement, it would be advisable to implement the actions described in the document "Steps for Radiation Safety in Pediatric Interventional Radiology" of Image Gently Alliance [29].

  1. In this manuscript, the included data is only the data of one hospital in Colombia. It is suggested to combine the relevant data of other hospitals for unified analysis and comparison, and increase the sample size, which will be more convincing and provide guiding significance.

Corresponding author answer:

Thanks for the comment. As indicated in the work, these initial results are from a single center in Colombia. Therefore, we have indicated that this is a limitation of the article along with other issues “Regarding the study’s limitations, it is a single-center study where, based on statistics, it was possible to establish a proposal of DRLs for diagnosis in diagnostic and therapeutic procedures for classification categories based on age and weight bands, despite having a low number of procedures. Another variable to consider was the manual collection of data, which may add some error while typing the information. To minimize the possibility of typing error, we validated the data by looking at the DICOM dose structured report for each procedure.”

We know of other centers in Colombia that participate in OPRIPALC, but their volume of data is too low to be able to do a job that allows us to propose national DRLs, however in the future that is our idea.

Round 2

Reviewer 1 Report

Your work is substantially improved.

You still missed some of my recomandations.

Paragraph 3 and 4 are the same.                                         

You still used the word band to describe your groups.

There are some additionally informations in a table's lagend that I believe you should write them after the table (the table in which you evaluate the literature about similar subjects).

In the disscution section you missed comparing your results to those from literature.
